# Lightweight Chassis Design of Hybrid Trucks Considering Multiple Road Conditions and Constraints †

**Shuvodeep De** [1,2,*], **Karanpreet Singh** [1], **Junhyeon Seo** [1] , **Rakesh K. Kapania** [1], **Erik Ostergaard** [3], **Nicholas Angelini** [3] and **Raymond Aguero** [3]

1    Kevin T. Crofton Department of Aerospace and Ocean Engineering, Virginia Tech, Blacksburg, VA 24061, USA; kasingh@vt.edu (K.S.); jhseo@vt.edu (J.S.); rkapania@vt.edu (R.K.K.)
2    Chemical and Biological Engineering, University of Alabama, Tuscaloosa, AL 35487, USA
3    Metalsa Roanoke, 8514, 184 Vista Dr, Roanoke, VA 24019, USA; erik.ostergaard@metalsa.com (E.O.); nicholas.angelini@metalsa.com (N.A.); raymond.aguero@metalsa.com (R.A.)
\*    Correspondence: shuvode@vt.edu
†    This paper is an extended version of our paper published in AIAA Scitech 2019 Forum, 7–11 January 2019, San Diego, CA 92101, USA.

**Abstract:** The paper describes a fully automated process to generate a shell-based finite element model of a large hybrid truck chassis to perform mass optimization considering multiple load cases and multiple constraints. A truck chassis consists of different parts that could be optimized using shape and size optimization. The cross members are represented by beams, and other components of the truck (batteries, engine, fuel tanks, etc.) are represented by appropriate point masses and are attached to the rail using multiple point constraints to create a mathematical model. Medium-fidelity finite element models are developed for front and rear suspensions and they are attached to the chassis using multiple point constraints, hence creating the finite element model of the complete truck. In the optimization problem, a set of five load conditions, each of which corresponds to a road event, is considered, and constraints are imposed on maximum allowable von Mises stress and the first vertical bending frequency. The structure is optimized by implementing the particle swarm optimization algorithm using parallel processing. A mass reduction of about 13.25% with respect to the baseline model is achieved.

**Keywords:** hybrid truck chassis; finite-element modeling; structural optimization; lightweight structure; stress computation



## 1. Introduction

Since their inception, the design of automobiles has changed considerably. The Department of Energy is currently investing millions of dollars in research and development of the generation of energy-efficient automobiles (https://www.energy.gov/articles/energy-department-announces-137-million-investment-commercial-and-passenger-vehicle). The energy efficiency of vehicles can be achieved by improving engine performance, hybridization, improving the aerodynamics, and making structural components lightweight.

Lightweight components not only make the vehicles more energy-efficient, but they also result in improvement in road performance and handling. In the past, the dimensions of automobile components were determined mostly by hand calculations by applying the principles of strength of materials. However, the last few decades have seen an exponential rise in computational power, which makes detailed structural analysis of complex structures possible using various numerical techniques. The finite element method is one of such numerical methods, which gained widespread popularity for structural analysis ever since the publication of the seminal paper by Turner et al. [1] and a series of papers published by Argyris and Kelsey [2], which subsequently appeared in the form of a book. The development of user-friendly computer-aided design (CAD) and finite element analysis (FEA) software

certainly made it possible to generate and analyse detailed three-dimensional (3D) modeling and perform analysis of complex structures. To automate the design process, finite element methods are usually integrated with numerical optimization algorithms. It is particularly important that the design satisfies all geometric and manufactural constraints. For extremely complex structures like those of large commercial vehicles, the modeling and analysis can still, despite enormous advances in both hardware and software, be quite expensive. In most industries, optimum structural dimensions and configurations are determined by engineering experience and trial-and-error. It requires a lot of human resources and ingenuity to generate and analyze numerous models before a design is finalized.

Multiple research groups worldwide are working on the design and manufacturing of lightweight vehicles, which can dramatically reduce the design cost. A great deal of research is going on developing new algorithms and techniques related to multidisciplinary and multiobjective optimization of automobile parts. Some of the popular areas of research are size/shape optimization [3], topology optimization, lattice-based optimization, etc. [4]. In size optimization, usually, the cross-sectional dimensions of structural components are considered as design variables. In shape optimization, the geometry of a component is defined by a set of parameters that can be varied. Topology optimization is one of the most modern methods of optimization where the density of elements of the FEA model is considered as a design variable while adding total volumetric fraction as a constraint. Various studies on automobile frame optimization (including shape/topology optimization considering multiple constraints) can be found in the work of Zuo et al. [5–11]. Miao et al. [12] developed a multidisciplinary design optimization framework for fatigue life prediction of automobiles.

Cavajzzuti et al. [13] used topology, topometry, and size optimization to design automotive chassis while satisfying the structural performance constraints as per Ferrari standards. The design, when compared to the commercial Ferrari F458 chassis, showed significant weight reduction. Wang et al. [14] studied the topology optimization approach for longitudinal beam shape frames with variable cross-sections to derive a reliable chassis design. They achieved the optimized frame, which was robust and had a low natural frequency. Kurdi et al. [15] compared diverse heavy-vehicle frames with different mass and torsional stiffness. The authors found an effective design with low weight and maximum torsional stiffness. Kang et al. [16] presented the optimal design of a heavy-vehicle by applying the analytical target cascading (ATC) methodology. They solved design problems for heavy-duty trucks and buses in the presence of a suspension system. Rajasekar et al. [17] applied the genetic algorithm to optimize the chassis with various rectangular cross-sections. Jin and Wang [18] performed the strength analysis of a simplified suspension model. The authors simplified the suspension with an equivalent beam to calculate the frame's strength under diverse load conditions.

Techniques like topology optimization are computationally expensive. It is reasonable to optimize small components using topology optimization. However, it is not practical for multidisciplinary design optimization of a complex structure like vehicle chassis involving multibody interaction. In problems involving complex load paths, topology optimization often results in designs infeasible to be manufactured by conventional manufacturing approaches. Even though developing a surrogate model is one way to tackle a problem involved in a highly complex structure, it requires the availability of optimal experimental designs (OED). Performing experiments or simulations can be an enormously expensive task. For complex structures, a more reasonable approach is to develop a simplified equivalent model that can represent the physics reasonably accurately.

The primary purpose of this work is to develop a computational framework for optimizing the structure of truck chassis using a mathematical model that is relatively accurate in representing the actual structure considering the stress, modal frequency, and manufactural constraints. The medium-fidelity model is verified with the detailed finite element model of the truck chassis considering stiffness and modal frequencies as metrics. As the disfeatured medium-fidelity model is likely to contain stress singularities (at sharp edges and points of beam attachments), the maximum von Mises constraint cannot be

considered a constraint. To circumvent this issue, we proposed a term called 'Violation' as the fraction of the total area over which von Mises stress is greater than the permissible value and constraint is imposed on its maximum value. As the vertical bending mode frequency has a significant effect on the performance and the passenger-comfort of the vehicle, a constraint is imposed on its minimum value. The framework incorporates the modal assurance criterion (MAC) to identify the first vertical bending model and the corresponding modal frequency to compute the constraint on vertical bending frequency. The framework, as developed, employs a general approach to performing structural optimization of a complex structure under stress and modal frequency for a specific mode shape. Although it was only done for the first bending mode, the approach based on modal assurance criteria could also be employed for other modes, e.g., a torsional mode.

Furthermore, an unconventional structure of the side rail of the truck chassis is explored using the optimization framework. It is a C-section but with a central drop and a rectangular top and a bottom plate attached. The shape of the profile is defined by a set of continuous design variables. The thickness of the top and bottom plate, the web, and flanges of the channel change along the length, and they are defined by another set of discrete variables. The geometry and mesh are generated using commercial FEA software, MSC. PATRAN (Version: 2014, MSC Software Corporation, Newport Beach, CA, USA) [19]. By the orthogonal method, a set of load conditions, each corresponding to a road event, is derived, and linear static analysis is run using MSC. NASTRAN [20]. Reaction forces from the road are applied at the wheel locations of the suspensions. The rail-shaped chassis with suspension is an unconstrained structure. To achieve a static equilibrium, the 'inertia relief' method is used. The aim of this work is to minimize the structural mass of the rails of a very large commercial truck chassis while satisfying multiple constraints and considering multiple load cases. The constraints include maximum allowable von Mises stress, minimum stiffness, and first vertical bending frequency. The metaheuristic optimization algorithm, particle swarm optimization (PSO) algorithm, is used for optimizing the design variables.

Overall, in this work, detailed geometry parameterization and integration of cross-members with the side-frame and verification of the medium-fidelity with the high fidelity model are described. The method for calculating the vertical bending stiffness and the influence of geometry on vertical bending frequency and stiffness and verification of the medium-fidelity assembly with high fidelity results are established. The integration of the suspensions and point masses to create a complete assembly with the method of detecting the vertical bending mode in an automated way is studied during the optimization process. The load cases for static analysis are established, and, finally, the optimization methodology is established.

## 2. Modeling the Side Rails

In the parametric model of the rail, the cross-section is an important design component. Fifteen continuous design variables define the web height and flange thicknesses in different regions. Figure 1 shows the top view and side view of the rail, and the dimensions that are labeled in red are considered to be variables. The variables *Rab1*, *Rab2*, *Rbc1* and *Rbc2* denote fillet radii. The terms FWW and RWW denote the "Front Wheel Width" and "Rear Wheel Width", respectively, and they are considered constant. All the dimensions cannot be varied independently as they are linked by the following set of equations:

$$
\begin{aligned}
Lab &= \frac{Hb - Ha}{\tan(Aab)} \\
Lbc &= \frac{Hb - Hc}{\tan(Abc)} \\
Lb &= TRL - La - Lab - Lbc - Lc
\end{aligned}
\tag{1}
$$

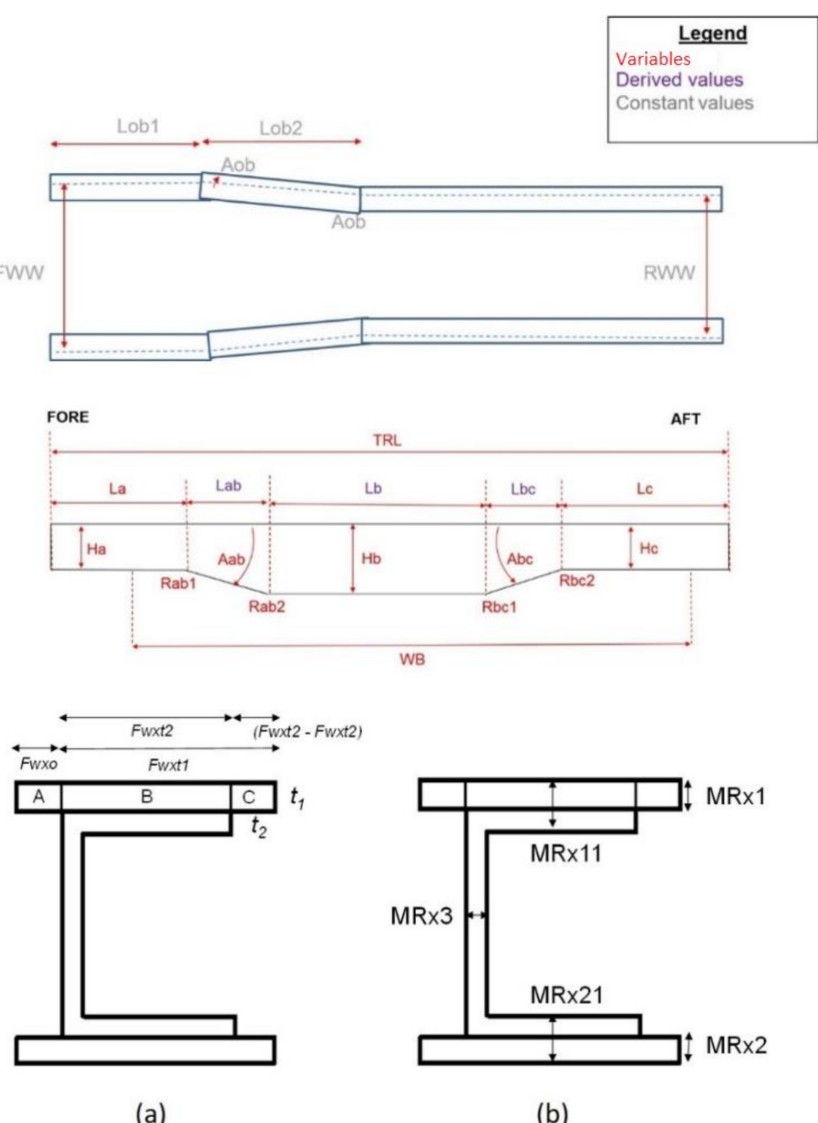

**Figure 1.** (**a**) Top view and side view of rail; (**b**) cross section of side rail (consisting of C-section with top and bottom plates).

The rail can be divided into several transverse sections, each characterized by an independent set of top and bottom plate dimensions and thicknesses. Sections are numbered in order (from front to the back), and Figure 1b shows the $x$th rail cross section; x being the section number. Figure 1b (i) shows the dimensions of the top plate, defined by variables *Fwxo*, *Fwxt*1 and *Fwxt*2. Figure 1b (ii) shows the variables specifying flange and rail thicknesses (*MRx*1, *MRx*11, *MRx*3, *MRx*21, MRx2). The range of the variables specifying the dimension needs to be consistent with manufacturing limitations and geometric constraints to ensure the rail does not overlap with the geometry of other components of the truck. The thickness design variables should be assigned only values of thicknesses of available grades of sheet metal in the manufacturing industry.

In this work, optimization was performed using a limited number of design variables. Variable *La* is considered to be the sum of *Lob*1 and *Lob*2, i.e.,

$$La = Lob1 + Lob2 \tag{2}$$

Further, the fillet radii were kept constant and equal to 1000 mm. The rails were divided into three sections (denoted as Sections 1–3) as shown in Figure 2. Each section was characterized by different thicknesses and dimensions of the top and bottom plates.

The dimensions of the top plate were specified by *Fwxt1*, *Fwxt2* and *Fwxot* (see Figure 1b) where '*x*' denotes Section # and '*t*' indicate '*top*'. Similarly, dimensions of the bottom plate were specified by *Fwxtb1*, *Fwxb2*, and *Fwxob* where '*b*' indicates '*bottom*.' The ratio of *Fwxt1* to *Fwxt2* was kept constant for each of the sections and denoted by $R_1 = Fwxt1/Fwxt2$. Similarly, $R_2$ was defined as $R_2 = Fwxb1/Fwxb2$.

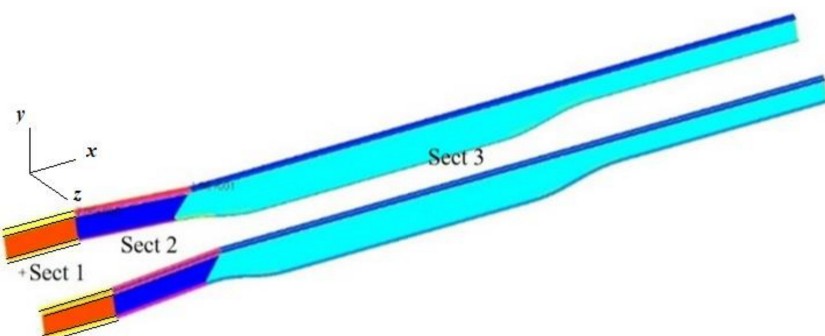

**Figure 2.** Rails divided into three sections with different thicknesses (shown using different colors).

Since there are three sections, 15 additional design variables were required to specify the thickness values. The thickness values are real numbers with appropriate ranges. The model is meshed with linear quadratic shell elements with a maximum edge length of 10 mm.

## 3. Cross Members' Integration and Complete Assembly

The two side rails were linked with a total of seven cross members. The cross members are represented in Figure 3a using beams with cross sections. They were attached to the side rail using multiple point constraints to create the medium-fidelity finite element model of the chassis, as shown in Figure 3b. Similar to the baseline design, the rails and the front three cross members were modeled using steel (Young's modulus = 200 GPa, Poisson ratio = 0.3, Density = 7900 kg/m$^3$) and the rest of the cross member using aluminum (Young's modulus = 73 GPa, Poisson ratio = 0.3, density = 2700 kg/m$^3$).

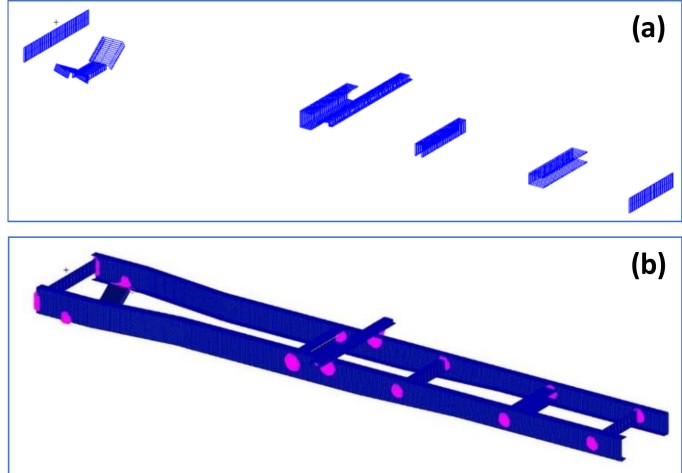

**Figure 3.** (**a**) Cross members and relative positions in the structure; (**b**) medium-fidelity finite element model of the baseline design.

## 4. Stiffness and Modal Frequency Calculation

The modal frequencies and vertical bending stiffness were used as metrics to verify the medium-fidelity finite element model. Figure 4a shows the approach for calculating the vertical bending stiffness of the frame. The boundary conditions were applied at the wheel locations, as shown in the figure. A load *F* of 1000 N was applied in the middle

of the chassis, and the maximum vertical deflection was computed using static analysis. The vertical bending stiffness was calculated as ($\delta_\mathbf{v}$ being the vertical displacement):

$$k_v = \frac{F}{\delta_v} \tag{3}$$

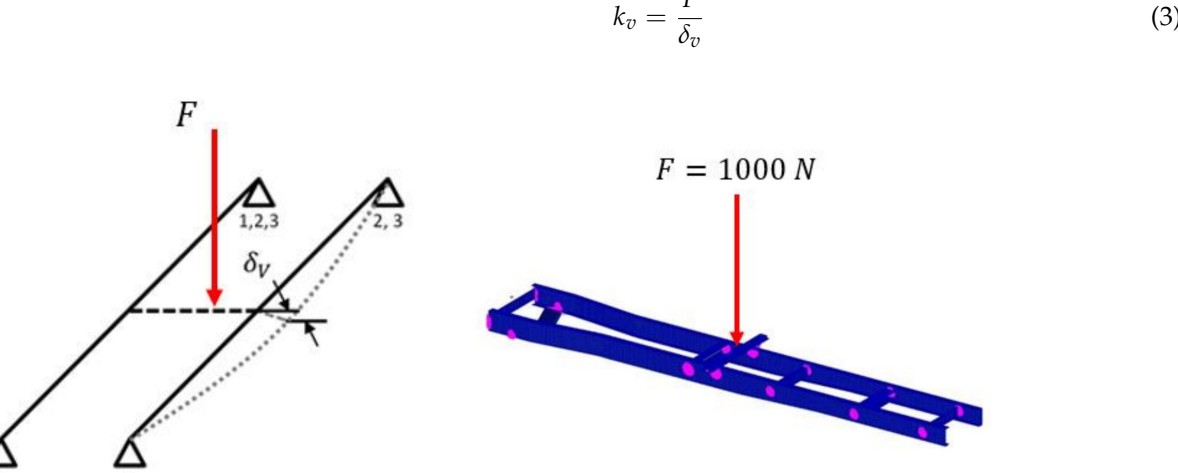

**(a)**                                                                 **(b)**

**Figure 4.** (**a**) Vertical bending deflection under applied load in the middle. (**b**) Finite Element Model with vertically downward force.

For the vertical bending stiffness calculation, the four mounting points were modeled as nodes connected to the upper and the lower flanges using multiple point constraints (MPC). The force *F* was applied in the vertically downward direction at a central node attached to the flange of the left and the right rail using MPCs, as shown in Figure 4b.

In order to gain an insight into the influence of the design variables on the first vertical bending frequency ($f_v$) and vertical bending stiffness ($k_v$), these values were obtained for a set of randomly generated designs and compared with the values corresponding to the baseline of truck chassis. Figure 5a,b show the plots of vertical stiffness vs. the mass of the randomly generated and the first vertical bending frequency vs. the mass for the same random designs, respectively. The design marked as '*Model of interest*', as shown in Figure 5c, had a higher stiffness compared to the baseline truck chassis yet had a significantly lower mass.

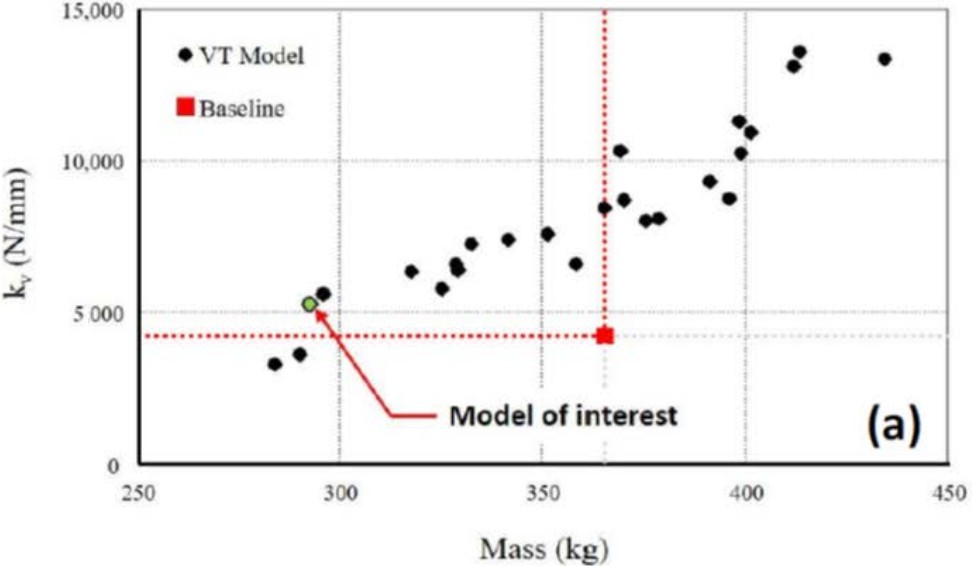

**Figure 5.** *Conts.*

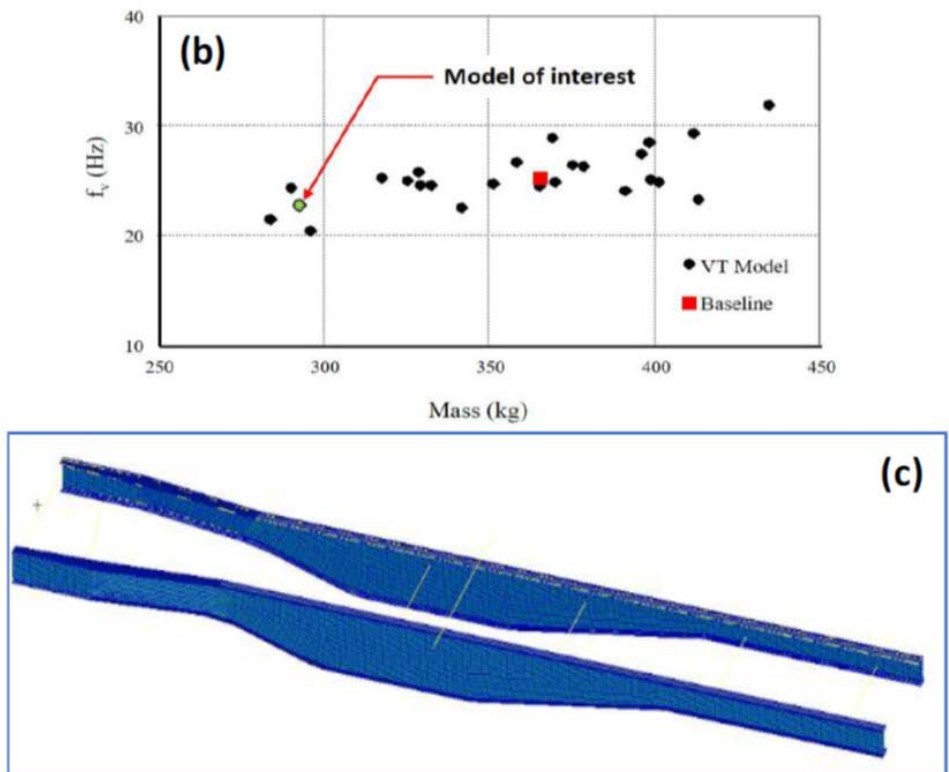

**Figure 5.** (**a**) Vertical bending stiffness vs. mass for random designs, (**b**) first vertical bending frequency vs. mass for random designs, and (**c**) the side rails of the model of interest (lower mass compared to baseline but with high vertical bending stiffness).

## 5. Model Verification

The stiffness and mass distribution of the medium-fidelity model of the baseline design (consisting of a no-drop section in the web) was verified by comparing the first torsional deformation frequency, first lateral bending frequency, first vertical bending frequency, and the vertical bending stiffness with those of a high-fidelity model of the baseline design of truck chassis shown in Figure 6. The high-fidelity model consisted of detailed models of the cross members (meshed with linear quadrilateral plate elements) and mountings (meshed with linear three-dimensional tetrahedral solid elements). It also accounted for the detailed geometric features of commercially used rails and the connecting brackets. It comprised a total of 269,562 nodes and 671,707 elements. Table 1 summarizes the frequencies and vertical bending values of the medium-fidelity model of the baseline design for various mesh sizes and the same corresponding to the high-fidelity model. The metrics of the medium-fidelity model and the high-fidelity model shows good agreement.

**Table 1.** Comparison of high-fidelity and medium-fidelity modal frequencies.

| Case | First Torsional Deformation Frequency (Hz) | First Lateral Bending Frequency (Hz) | First Vertical Bending Frequency (Hz) | Vertical Bending Stiffness (N/mm) | Violation Value |
|---|---|---|---|---|---|
| High-Fidelity Model | 3.45 | 12.57 | 24.26 | 4027 | N/A |
| Shell Element (Element Size 12.5 mm) | 3.29 | 10.64 | 25.34 | 4168 | 0.010944 |
| Shell Element (Element Size 15 mm) | 3.27 | 10.58 | 25.16 | 4182 | 0.01012 |
| Shell Element (Element Size 17.5 mm) | 3.21 | 10.57 | 25.31 | 4154 | 0.00991 |
| Shell Element (Element Size 20 mm) | 3.21 | 10.54 | 25.121 | 4125 | 0.009498 |

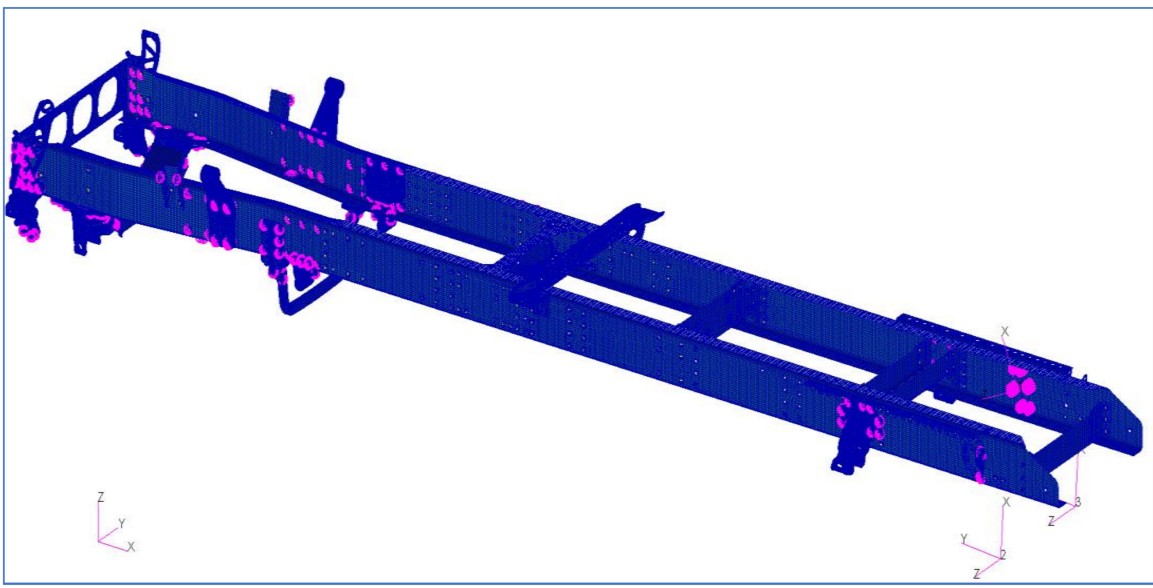

**Figure 6.** High-fidelity model of the baseline design.

Furthermore, it was found that, with an increase in the element size, the model reported a lower value of both vertical bending stiffness and modal frequencies. This can be accounted for because a smaller number of MPCs were created with a decrease in the number of nodes, which caused a drop in the structural stiffness. Figure 7 shows the first torsional, first lateral bending, and first vertical bending mode of the structure.

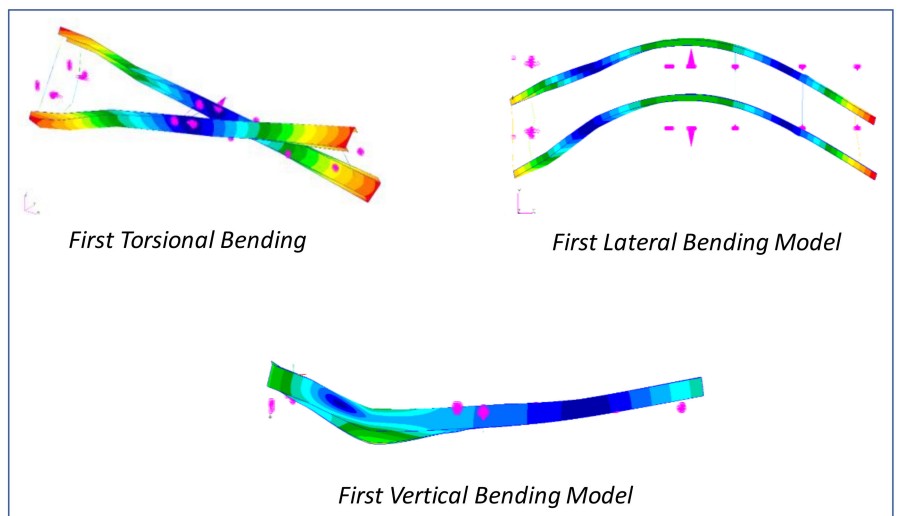

*First Torsional Bending*          *First Lateral Bending Model*

*First Vertical Bending Model*

**Figure 7.** First torsional bending, first lateral bending, and first vertical bending modal shapes.

## 6. Suspension Integration

In the current work, parameterization and optimization were conducted only on the chassis. In order to transfer the loads from the road to the frame, a simplified model of the front and the rear suspensions, as shown in Figure 8a,b, respectively, was created by Metalsa. The front suspension was similar to Hendrickson AIRTEK NXT front air suspension (https://www.hendrickson-intl.com/Truck/On-Highway/AIRTEK-NXT). The rear suspension was similar to the Hendrickson HTB LT (https://www.hendrickson-intl.com/Truck/On-Highway/HTB-LT), but for a 4 × 2 vehicle layout. CAD was provided to define the kinematic hard point of the suspension, the brackets, the spring hanger geometry, and interfaces to the frame. Suspension radial and cylindrical bushing stiffness, as well as the air spring vertical

stiffness, were obtained from Original Equipment Manufacturer (OEM) datasheets. The front suspension leaf spring stiffness was adjusted to match the bulk vertical stiffness measured on the physical vehicle. Several 'ride heights' (distance from the suspension bump stop to the frame rail) and 'wheel loads' (force at the wheel in vertical direction were measured under different payload levels to develop an experimental target stiffness.

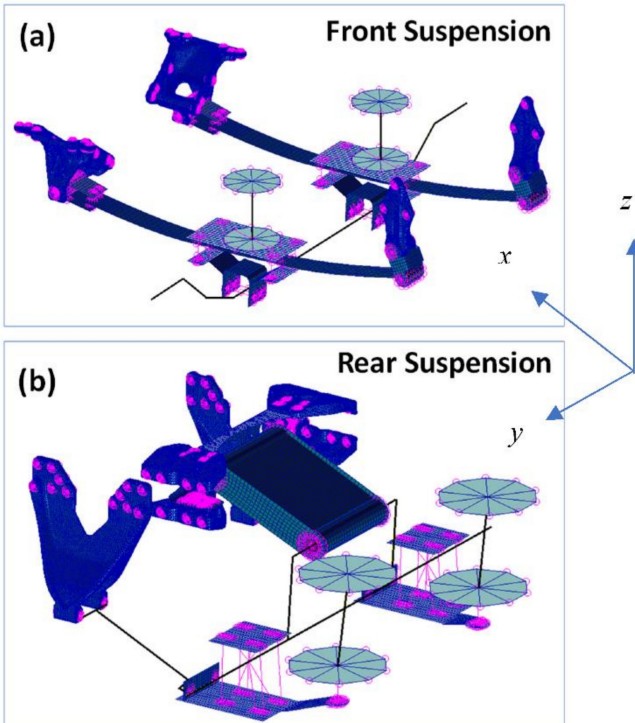

**Figure 8.** Finite element model showing the (**a**) front and (**b**) rear suspensions.

In a typical truck chassis, the side-rails and cross members are connected by bolted joints. Detailed analysis of bolted joints is computationally demanding as it involves contact mechanics with several mating surfaces. That is why, in this work, a simplified equivalent represented the bolted joints. The joint was modeled using a rigid bar element (for the bolt) and multiple point constraints (MPCs). MPCs are essentially a set of rigid bars that connect a node to multiple nodes of a surface mesh. A rigid bar element was created across the center location of the boltholes on the two connected plates. MPCs were created between the nodes at the periphery of the bolthole and the center node (ending nodes) of the bar element. In these connections, all the degrees of freedom of the boundary nodes were constrained to be dependent on the center node. Figure 9 shows a typical bolthole in the model with MPCs.

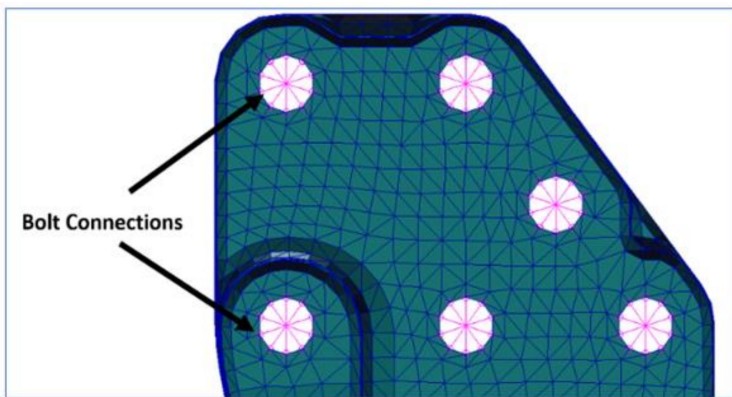

**Figure 9.** Simplification of bolted joints.

Furthermore, in the current approach, a geometric constraint was added such that the top of air-springs in the suspensions touched the bottom flange of the side rails. Finally, MSC.NASTRAN input files for applied forces based on the load cases were imported and applied at the required nodes, and static analysis was carried out to calculate the stresses and displacements.

Figure 10 shows the complete shell element-based representation of the truck chassis as created by the integration of the side frames, front suspension, rear suspension, and point masses representing the center of gravity of the engine, including the air-tank and other features which were not considered for optimization in this problem.

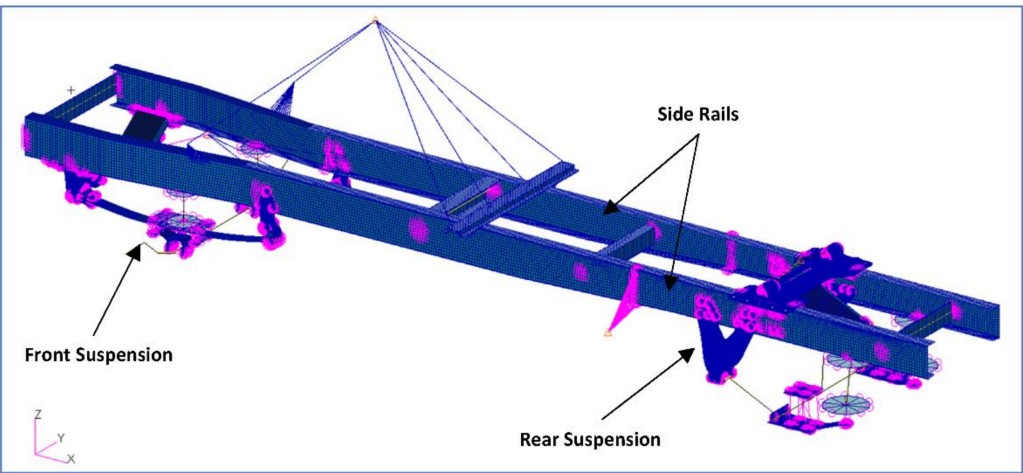

**Figure 10.** Creation of complete model (including suspensions and point masses).

## 7. Static Analysis

In this research, multiple load conditions were considered for static analysis. The following extreme five road events were considered to assume the behavior of proving ground tests:

(i)     Both front wheels in bump event;
(ii)    Both rear wheels in bump event;
(iii)   Both front tires in pothole event;
(iv)   Both rear tires in pothole event;
(v)    Maximum breaking condition.

The loads on four wheels regarding those five road events were created under the assumption of orthogonal load cases. Details on the construction of these load cases can be found in the paper by Ostergaard et al. [21]

It was assumed that all load conditions would be somewhere in between the above-mentioned cases. For each of the load cases, linear static analysis was conducted using the inertia-relief method [22]. Inertia-relief is a popular method of analysis for unconstrained moving structures. Nelson et al. [23] used the inertia relief analysis to estimate the impact of loads on the space structure. Morton et al. [24] applied this method to calculate the distribution of flight load on an unconstrained helicopter rotor. Vallejo et al. [25] simulated a finite element model using inertia relief to predict the fatigue behavior of a heavy truck chassis. Pagaldipti et al. [26] studied the influence of inertia relief on optimal designs. Saito et al. [27] carried out full automobile optimization procedures with the inertia relief analysis. Zhang et al. [28] used the inertia relief option to perform stress analysis on a mine dump truck frame and proposed essential elements for the optimization of a commercial vehicle. Table 2 shows the g-forces on wheels corresponding to the assumed road conditions (RC). The *X*-axis is in the direction of the forward motion of the vehicle while the *Z*-axis is normal to the road.

**Table 2.** Load cases using the method of superposition (numbers indicate g-force magnitudes).

| Wheel -> | Front Left | | | Front Right | | | Back Left | | | Back Right | | |
|---|---|---|---|---|---|---|---|---|---|---|---|---|
| Direction -> | X | Y | Z | X | Y | Z | X | Y | Z | X | Y | Z |
| Front Both Bump | 0 | 0 | 1.75 | 0 | 0 | 1.75 | 0 | 0 | 1 | 0 | 0 | 1 |
| Rear Both Bump | 0 | 0 | 1 | 0 | 0 | 1 | 0 | 0 | 1.75 | 0 | 0 | 1.75 |
| Front Both Pot Hole | 0.75 | 0 | 1.75 | 0.75 | 0 | 1.75 | 0 | 0 | 1 | 0 | 0 | 1 |
| Rear Both Pot Hole | 0 | 0 | 1 | 0 | 0 | 1 | 0.75 | 0 | 1.75 | 0.75 | 0 | 1.75 |
| G-Stop Forward | 0.4 | 0 | 1 | 0.4 | 0 | 1 | 0.4 | 0 | 1 | 0.4 | 0 | 1 |

The constraints included maximum allowable von Mises stress and minimum first vertical bending frequency. The defeatured finite element model shown in Figure 10 contains several sharp edges and places where beams are attached to surfaces. These are the result of simplification, and such features do not exist in the real structures. However, a simple static analysis of these medium-fidelity models often shows stress singularities around these areas. The stress value here was significantly higher than elsewhere [29–31]. For stress-based optimization, usually, these regions need to be excluded. To do so, we defined a parameter entitled '*Violation*' as

$$Violation = \frac{Shell\ Area\ where\ \sigma_{vMises} > \sigma_{allowed}}{Total\ Surface\ Area} \quad (4)$$

Instead of imposing a constraint on the maximum von Mises stress in the system, the constraint was imposed on maximum 'Violation.' For the model with no stress singularities, the value of 'Violation' for the optimized design should be 0.

## 8. Mode Detection

In the optimization problem, the vertical bending natural frequency of the truck frame was added as a constraint, and it needs to be greater than 20 Hz. The first step is obviously to run the free vibration analysis, and it was performed using MSC.NASTRAN.

To find the frequency of the vertical bending mode from a set of all free vibration modes of the truck chassis design, modal assurance criteria (MAC) were implemented [32]. If there are two normalized eigenvectors: $\{\Phi_A\}$ and $\{\Phi_B\}$, the MAC is defined as

$$MAC = \frac{\left|\{\Phi_A\}^T\{\Phi_B\}\right|^2}{(\{\Phi_A\}^T\{\Phi_A\})(\{\Phi_B\}^T\{\Phi_B\})} \quad (5)$$

The value of MAC is bounded between the values 0 and 1. The value 0 indicates that the two eigenvectors are completely orthogonal to each other. However, value 1 indicates that the two modes are fully matched. In this work, a reference eigenvector exhibiting vertical-bending deformation was taken, and for each of the vibration modes of a given design, the MAC was calculated. The mode with the highest value of MAC was considered as the vertical bending mode.

MAC can be calculated only when $\{\Phi_A\}$ and $\{\Phi_B\}$ are of the same dimension, i.e., the eigenvectors of the given design need to be of the same dimension as that of the reference eigenvector. This is almost impossible since the finite element models of different designs contain different numbers of elements. To resolve this issue, the displacement field of the eigenvector of the given design was interpolated on the finite element grid of the reference design to produce a new eigenvector which will be of the same dimension as of the reference eigenvector.

Figure 11 and Table 3 show an example implementation of the implemented procedure. It can be seen in Table 3 that the modal frequency of 31.94 Hz was the vertical bending natural frequency.

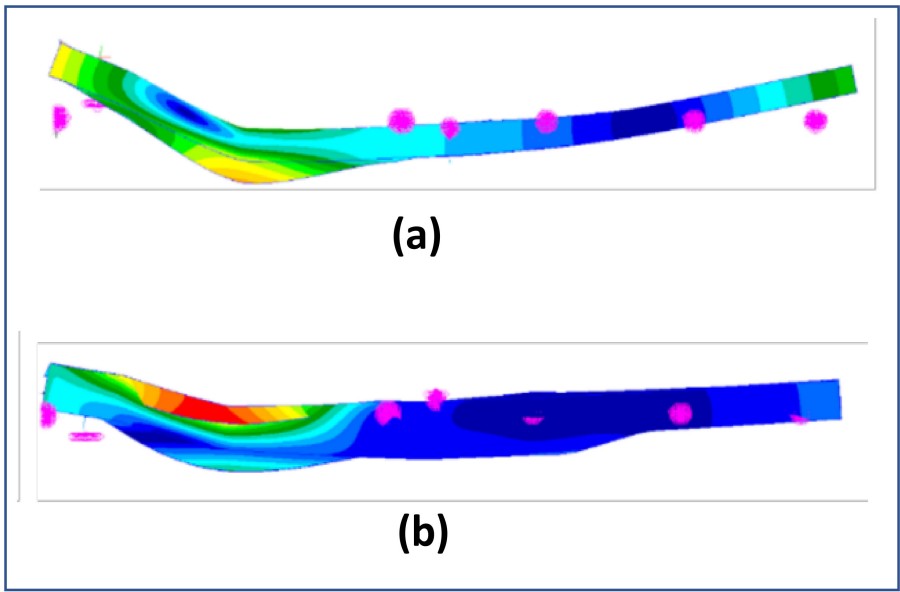

**Figure 11.** (**a**) Reference eigenvector (vertical bending mode) and (**b**) matching eigenvector from a set of test eigenvectors.

**Table 3.** Modal assurance criteria (MAC) values of test eigenvectors with respect to reference eigenvectors (see Figure 4b).

| Natural Frequencies (Hz) | Normalized MAC Value |
|---|---|
| 3.969 | $8.80 \times 10^{-8}$ |
| 10.57 | $1.22 \times 10^{-6}$ |
| 22.23 | $1.10 \times 10^{-4}$ |
| 25.43 | $2.23 \times 10^{-1}$ |
| 26.14 | $5.00 \times 10^{-3}$ |
| 27.31 | $5.68 \times 10^{-5}$ |
| 30.08 | $8.23 \times 10^{-5}$ |
| 31.94 | 1.00 |
| 33.63 | $7.50 \times 10^{-3}$ |

## 9. Optimization Framework

The aim was to minimize the structural mass of the rails while satisfying multiple constraints. Considering maximum 'Violation' to be 1%, the minimum value of first vertical bending frequency to be 20 Hz, and minimum vertical bending stiffness equal to that of the baseline truck model, the optimization problem can be mathematically written as:

$$\min(\text{Obj})$$

where,

$$\text{Obj} = W + 10^6 \left( \sum \max(0, g_i) \right)$$
$$g_1 = \frac{\max(Violation)}{0.01} - 1 \tag{6}$$
$$g_2 = \frac{f_v}{20} - 1$$

where $W$ is the structural mass of the rail, and $fv$ is the vertical bending frequency.

In this optimization problem, we dealt with structural weight in the range of $10^2$–$10^3$ kg. Hence, if the constraints are not satisfied, the objective function assigns a value ~$10^6$ kg and thus becomes an undesirable design.

The optimization was performed using a modified version of the Particle Swarm Optimization (PSO) algorithm, which is a heuristic optimization process and does not include the calculation of gradient. An explanation of this algorithm is given in the article by Kennedy et al. [33]. In every iteration, random particles (points in the design space) were distributed and evaluated. The particle's direction and position during the optimization process were updated (after the k$^{th}$ iteration) using Equations (7) and (8), respectively.

$$v_{k+1}^i = w v_k^i + c_1 r_1 \frac{\left(p^i - x_k^i\right)}{\Delta t} + c_2 r_2 \frac{\left(p_k^g - x_k^i\right)}{\Delta t} \tag{7}$$

$$x_{k+1}^i = x_k^i + v_{k+1}^i \Delta t \tag{8}$$

where $x_k^i$ are the design variables and are called the positions of the particles, $v_k^i$ is the velocity of the particle, which is used to update the position; $r_1$ and $r_2$ are the uniform random numbers between 0 and 1; $c_1$ and $c_2$ are known as thrust parameters; $w$ is the inertia weighting parameter of velocity; $p^i$ and $p_k^g$ are the best particle position (throughout iterative history) and the best swarm position, respectively. In Equation (7) the second term is known as "individual correction" because $(p^i - x_k^i)$ is essentially the difference between the particle's current position and the best position in history. Thus, if the term increases, the particle is attracted more towards the best position. The third term in Equation (7) is called "social correction" as $\left(p_k^g - x_k^i\right)$ is the difference between the particle's position and the best position in the entire swarm, and hence it attracts the particle to the global best. The inertia weight parameter, $w$, decides the influence of the particle's velocity compared to the personal and social influences, and it decides the optimization convergence rate. The parameter $\Delta t$ is called the time step and is often taken as 1. The values of the parameters as considered in this work are listed in Table 4.

**Table 4.** Particle swarm optimization (PSO) parameters used in the present study.

| PSO Parameter | Value |
|:---:|:---:|
| $w$ | 0.78 |
| $c_1$ | 2 |
| $c_2$ | 2 |
| $\Delta t$ | 1 |

Convergence is said to have been achieved when the difference in the objective value for the particles in the swarm falls within a specified limit, or the maximum allowable number of iterations is reached. It is always recommended that, for PSO, if the convergence rate is too high, there is a higher chance that the search will end in a local optimum. Thus, it is always recommended not to use a too high value for $w$. The framework for the implementation of the algorithm is given in Figure 12.

A significant advantage of the PSO algorithm is that the computation of objective functions for each of the particles is independent. Hence, the algorithm can be parallelized easily (using the message passing interface, MPI). Further, the analyzed models can be stored in a database, which can be used by the industry for other studies requiring a large number of models of different specifications. For such studies, the manual development of models can be very cumbersome.

However, the classical PSO algorithm needed some modification in order to be implemented in our problem. Firstly, the computation of the objective function involving mesh generation and finite element analysis is computationally expensive, hence there was a chance of memory saturation, especially while running in a cluster shared by multiple users.

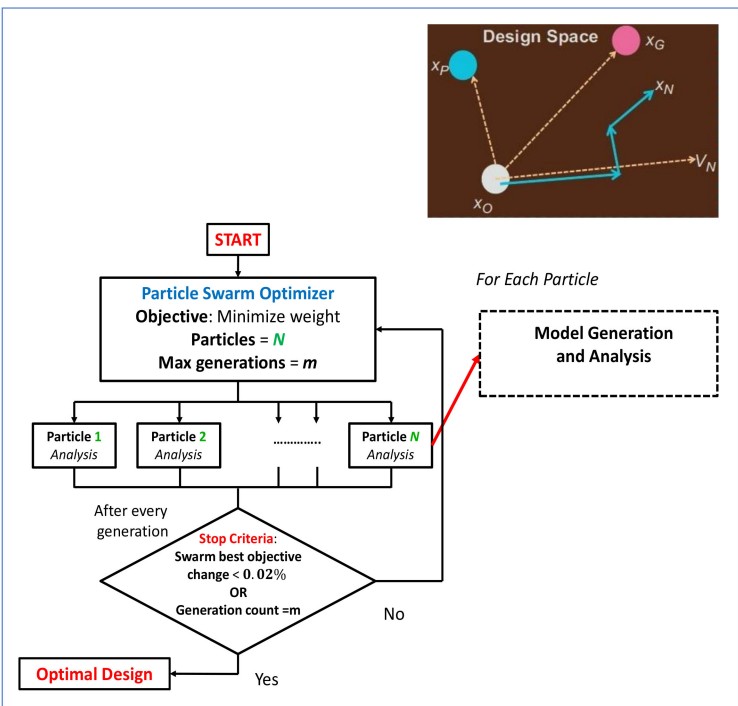

**Figure 12.** PSO algorithm applied to the weight minimization problem.

Secondly, while Virginia Tech has a limited number of licenses for commercial software MSC.PATRAN and MSC.NASTRAN, which are used in this research, there was a possible unavailability of the required number of licenses while running the optimization process using parallel processing. The optimization process was saved from stopping during unavailability of licenses and memory saturation by implementing the license cycle-check method and memory self-adjustment method [34,35] developed at Virginia Tech.

Moreover, for a certain set of design variables, MSC.PATRAN can fail to create the complete geometry, leading to analysis failure. When this happens, the objective function cannot be computed, and hence the algorithm fails to proceed. To prevent the optimization from stopping, a large value ($10^5$) was assigned to the objective function corresponding to it. This causes the optimizer to consider the design to be infeasible and thus the particle is discarded from the search space.

In order to perform optimization using the PSO algorithm, the model and mesh generation for different design variables, structural analysis, and evaluation of the constraints need to be automated. In this work, this automation was carried out using a python script. Once the constraints and hence objective function were evaluated for each of the particles, they were used as input to the PSO algorithm, which found the set of particles for the next iteration. The automated determination of the objective for the PSO algorithm is shown in Figure 13. The ranges for the shape design variables were set according to manufacturing limitations and those of the size design variables (representing thickness) according to the grades of sheet metal available.

Each "particle" corresponded to the generation of the finite-element model according to a set of design variables and running multiple types of structural analysis (modal analysis and vertical bending stiffness analysis) on the chassis and finally static analysis on the assembly for the five given load cases to calculate the maximum value of the 'Violation' factor.

For each of the load conditions, linear static analysis was performed using MSC.NASTRAN and the 'Violation' was calculated, as shown in Figure 14. Optimization could be performed considering the maximum value of 'Violation' or assigning different weights to 'Violation' corresponding to each load condition.

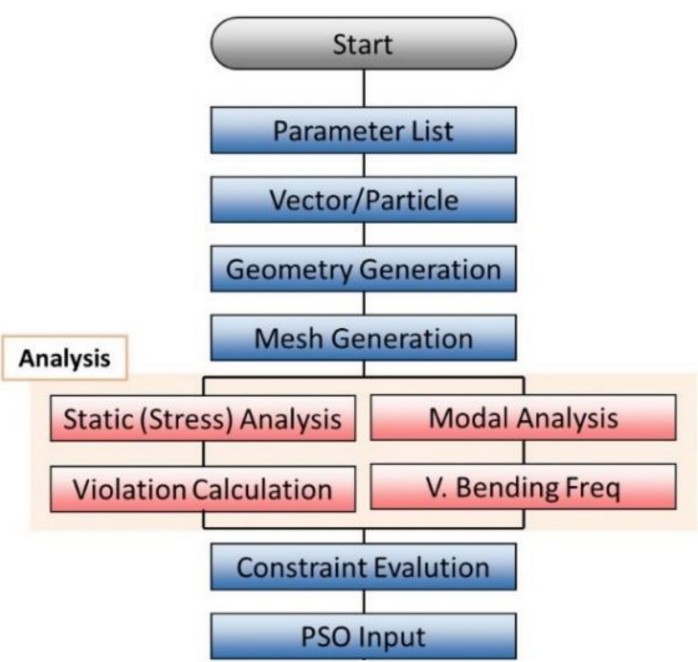

**Figure 13.** Python script for analysis and optimization.

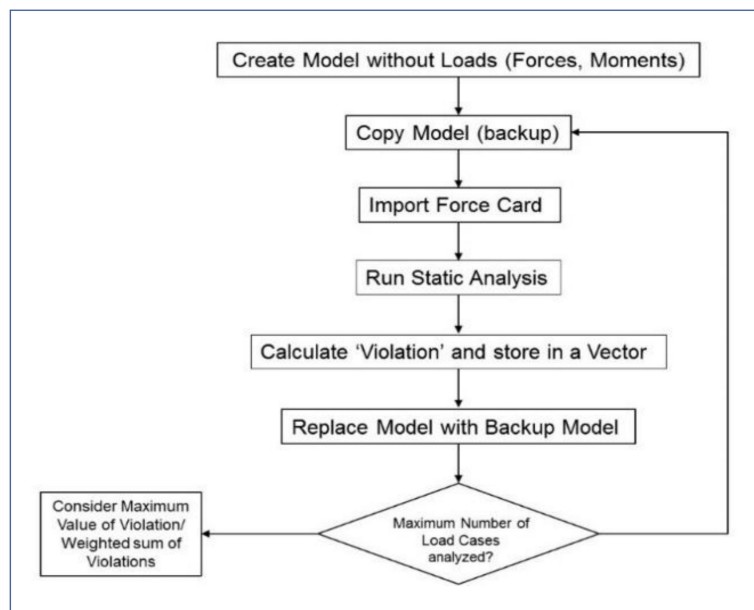

**Figure 14.** Process of calculating 'Violation' for multiple load cases.

The objective function was finally calculated. The objective function was set up such that it was equal to the structural mass only if all the constraints were satisfied. Otherwise, it took a very large value. The optimizer automatically considered the design as infeasible and tended to move away from it.

## 10. Optimization Results and Discussion

The optimization was run on a cluster having 48 cores with a clock speed of 2.2 GHz and a total RAM of 132 GB. Fifteen design variables defining the shape and 15 design variables defining the thicknesses were considered in the optimization. Sixty-six particles per iteration were checked, and the objective function was updated using the PSO algorithm. The optimization was run for a total of 15 iterations. It was found that the objective function remained unchanged after the first five iterations. Figure 15 shows the iteration

history. The best feasible design reported by the optimizer had a structural mass (without suspensions and point masses) of 275 kg, which is 13.25% less than the mass of the baseline design. The values of the vertical bending frequency and maximum 'Violation' factor corresponding to the best design were 20.5 Hz and 0.0086, respectively. While it was not possible to prove if a global minimum had been reached for such type of large-scale multivariable optimization problem, the fact that these values are close to the constraints and gives the confidence that the solution is close to the global minimum.

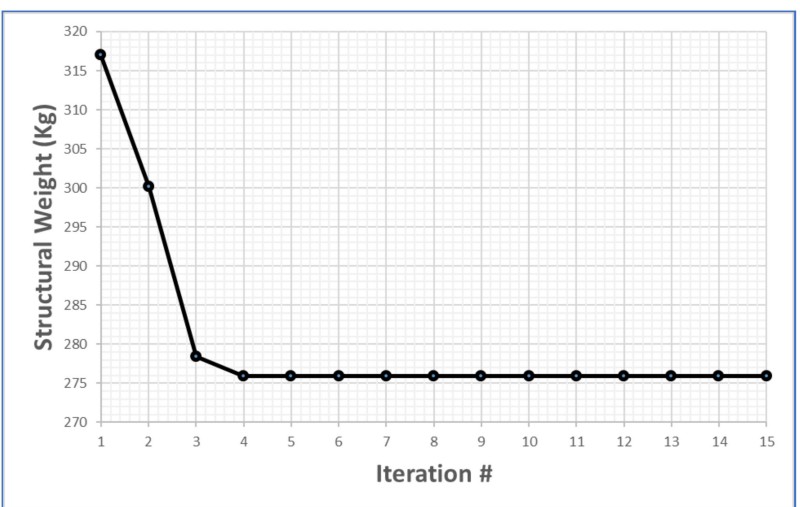

**Figure 15.** Iteration history for particle swarm optimization.

Figure 16a,b shows the vertical bending frequency vs. mass and violation vs. mass, respectively, for all the designs analyzed during the optimization. In these charts, the baseline design is indicated by the red dot, while the optimized design is indicated by a green dot.

Figure 17 shows the optimized design and thickness distribution for the side rails. The von Mises stress plots for this design in several events (event #1–5) are shown in Figure 18. As the design was guided by minimum gage thickness, it consisted of many low-stress zones. On the other hand, a high value of stress was found around regions where point masses and suspension leaf-springs were attached. As mentioned before, such stress 'hotspots' were expected in the medium-fidelity model due to the simplification of joints using MPCs and beam assembly. The method of optimization using the stress 'Violation' parameter (where a violation of stress constraint was allowed over a limited region) helped to arrive at a reasonable solution using a medium-fidelity representation of complex structures, like the truck chassis, which was analyzed in this research. Table 5 shows the influence of optimization on the first bending frequency, structural rigidity, and static strength.

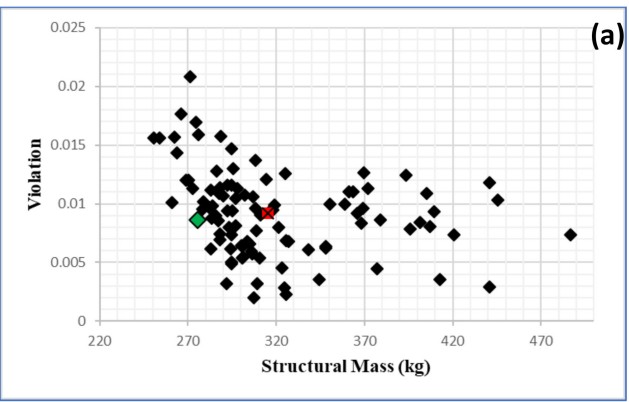

**Figure 16.** *Conts.*

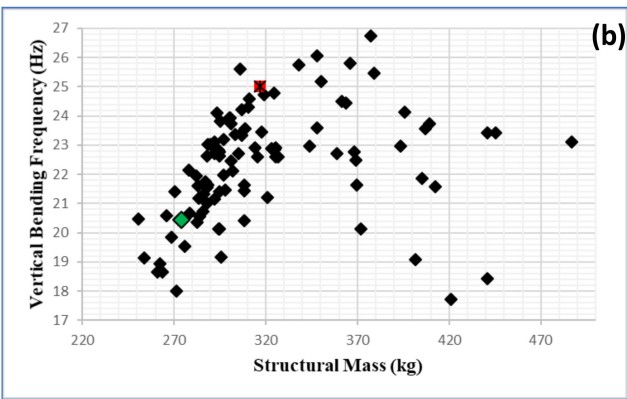

**Figure 16.** (**a**) Vertical bending frequency vs. structural mass for all designs analyzed and (**b**) violation vs. structural mass for all designs analyzed during optimization.

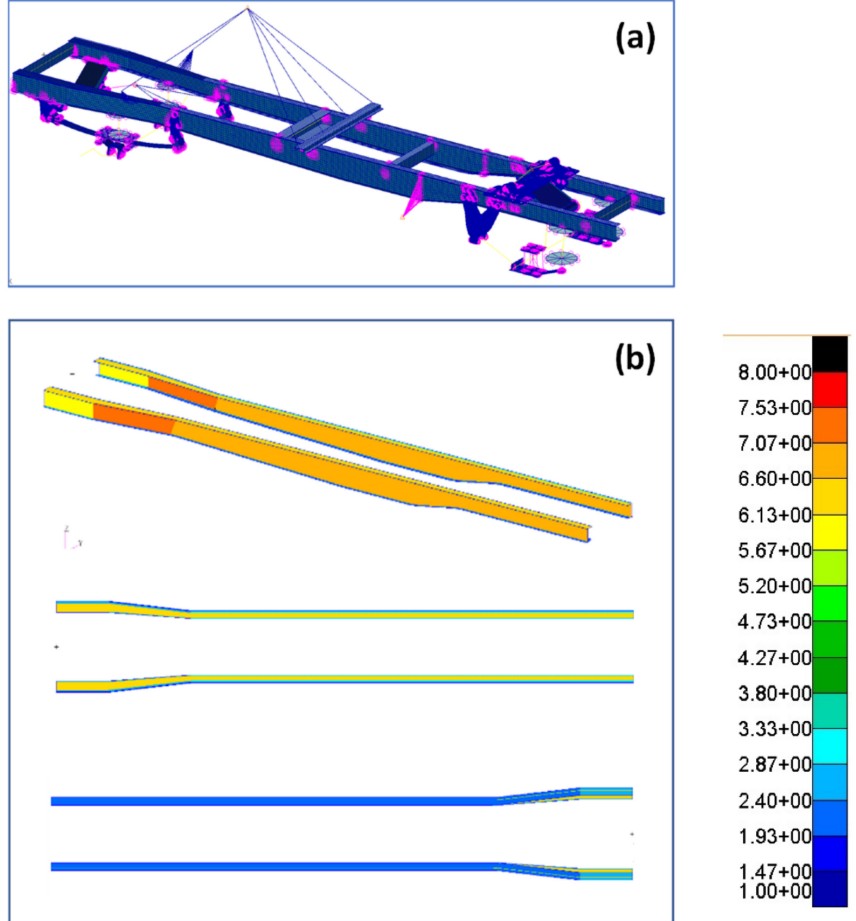

**Figure 17.** (**a**) Optimized design (mass of chassis without suspensions and point masses = 275 kg) (**b**) thickness distribution of the side rails.

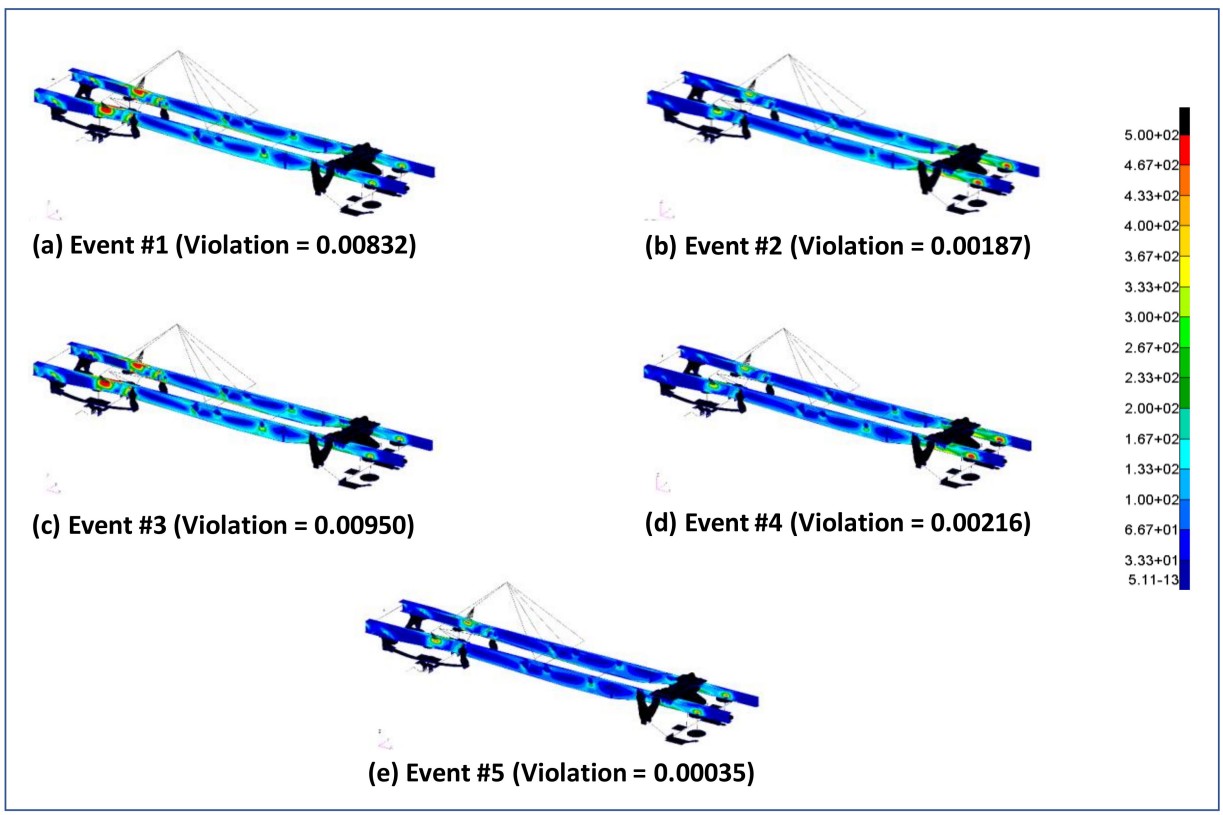

**Figure 18.** Von Mises stress distribution of optimized design for (**a**) Event #1, (**b**) Event #2, (**c**) Event #3, (**d**) Event #4 and (**e**) Event #5.

**Table 5.** Comparison between baseline and optimized model.

| Properties | Baseline Model | Optimized Model |
|:---:|:---:|:---:|
| Mass (Kg) | 317 | 275 |
| First Bending Frequency (Hz) | 25.12 | 20.5 |
| Bending Rigidity (N/mm) | 4125 | 268 |
| Violation Parameter (Stress) | 0.0094 | 0.0086 |

## 11. Conclusions

The article describes the parameterization of the side-rails for truck chassis by a large number of design variables and optimization considering several constraints, including maximum stress and minimum frequency of first vertical bending mode. A python script was developed, which automatically generated the geometry and mesh of the side-rails, integrated the suspensions, and points masses to create the simplified finite element model of the truck chassis. Normal mode analysis and static analysis for multiple load cases were performed on the entire model to evaluate the constraints in the optimization problem. The particle swarm optimization (PSO) algorithm was used to optimize the design variables to minimize mass while satisfying constraints. A mass reduction of 13.25% with respect to the baseline model is achieved. However, it was possible to go even further by applying topology optimization techniques to the configuration shown in Figure 17. The material can be removed from the side rails and the side rail mountings. Such a process will be challenging as manufacturing constraints need to be taken into account. This is something to be considered in future research.

**Author Contributions:** S.D., under the guidance of R.K.K. who also provided leadership towards developing the conceptual framework and task completion, developed the optimization framework and the finite element model of the chassis and obtained the final results. K.S. and J.S. contributed to the assembly of the finite element models of the components. E.O., N.A., and R.A. developed the high-fidelity finite-element model of the chassis, the front, and back suspensions and validated them. S.D. and R.K.K. prepared and revised the manuscript, respectively. They did this with assistance from J.S. All authors have read and agreed to the published version of the manuscript.

**Funding:** This research is partly funded by Metalsa (Contract Number: *AT-38971*).

**Institutional Review Board Statement:** Not applicable.

**Informed Consent Statement:** Not applicable.

**Data Availability Statement:** No new data were created or analyzed in this study. Data sharing is not applicable to this article.

**Acknowledgments:** We are thankful to Maria Eugenia Rodriguez Cantu and Francisco Gonzalez of Metalsa for their sincere collaboration.

**Conflicts of Interest:** The authors declare no conflict of interest.

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
