# Peer review of "Lightweight Chassis Design of Hybrid Trucks Considering Multiple Road Conditions and Constraints"

_wevj, doi:10.3390/wevj12010003_

Round 1

Reviewer 1 Report

The research significance of the article:

The highlight of this paper is to optimize the mass of the chassis by establishing a medium-fidelity finite element model and considering the working conditions under different road conditions to achieve the purpose of lightweight. The relevant processing methods in the finite element modeling process are relatively reasonable, and the optimization problem design meets the needs of practical applications.

The article needs major improvement

  • In the introduction part, there is no systematic analysis of the current research status and some deficiencies in existing research. It is recommended to supplement the latest literature on some optimization algorithms in structural lightweight design, multidisciplinary optimization, and multi-objective optimization design under various typical load conditions in recent years.

Such as: Miao B R, Luo Y X, Peng Q M, et al. Multidisciplinary design optimization of lightweight carbody for fatigue assessment[J]. Materials & Design, 2020, 194: 108910.

  • In the Section 7:Static analysis, it should be able to provide the analysis basis or standard for setting the load condition.
  • It is recommended to use a table to illustrate the comparative analysis of the influence of the optimized chassis' static strength, structural rigidity and first-order bending frequency.
  • Some details. Figure 4b is mentioned on page 6, but this figure is not given in the paper. The variables in some equations become "?", such as equations (4) and (5), these variables need to be represented correctly. There are other formulas, such as formula (6), (7) and (8), the display is not very clear.

Conclusion:

It is recommended that the paper undergo a major revision.

Author Response

We are truly thankful to the reviewer for reading our manuscripts and providing constructive remarks. Please find the attached.

Reviewer 2 Report

The topic of the paper falls within the scope of the journal. The shape and size optimizations were used to achieve the lightweight chassis design of hybrid trucks under the multiple road conditions and constraints. The authors are encouraged to deal with the following comments before the manuscript could be accepted:

  1. The current studies of the lightweight design for automobile body should be fully discussed in the introduction, some related papers such as:

Analytical sensitivity analysis method of cross-sectional shape for thin-walled automobile frame considering global performances. International Journal of Automotive Technology, online.

Shape optimization of thin-walled cross section for automobile body considering stamping cost, manufacturability and structural stiffness. International Journal of Automotive Technology, 21(2): 503-512.

Lightweight design of bus frames from multi-material topology optimization to cross-sectional size optimization. Engineering Optimization, 51 (6): 961-977.

Cross-sectional shape optimization for thin-walled beam crashworthiness with stamping constraints using genetic algorithm. International Journal of Vehicle Design, 73 (1-3): 76-95.

Stress sensitivity analysis and optimization of automobile body frame consisting of rectangular tubes. International Journal of Automotive Technology, 17 (5): 843–851.

Bi-level optimization for the cross-sectional shape of a thin-walled car body frame with static stiffness and dynamic frequency stiffness constraints. Proceedings of the Institution of Mechanical Engineers, Part D: Journal of Automobile Engineering, 229 (8): 1046–1059.

Component sensitivity analysis of conceptual vehicle body for lightweight design under static and dynamic stiffness demands. International Journal of Vehicle Design, 66 (2): 107-123.

should be discussed in the manuscript.

  1. The format of texts in each picture should be uniform, please check each picture in the manuscript.

  1. In Figure 1, the meanings of the texts “FWW” and “RWW” should be explained in the manuscript.

  1. In Section 4, the Figure 4b cannot be found in this manuscript, the number of the pictures should be reordered.

  1. In Section 6, the front and the rear suspensions were created to transfer the loads from the road to the frame, the details of the work processes of the suspensions should be introduced in the manuscript.

  1. The different parts of the truck chassis like side frames, front suspension and rear suspension should be marked in the Figure 10.

Author Response

We are truly thankful to the reviewer for reading our manuscripts and providing constructive remarks. Please find the attached file.

Round 2

Reviewer 1 Report

The paper has been carefully revised by the author, and the overall level has been well improved. After careful review, it was agreed to be published. It is recommended that the author carefully review the diagrams and texts, grammar and expressions of the paper, and publish them directly after modification.

Reviewer 2 Report

This manuscript has been improved greatly and all the comments are responded completely, so I recommend to accept this manuscript.